# Toward Resilient Watermark Detection: Stability-Aware Statistical Features for Machine-Generated Text

## Abstract

The widespread adoption of large language models (LLMs) has intensified the demand for principled methods to distinguish human- from machine-generated text. Watermarking provides a promising avenue, yet existing detectors exhibit sharp performance deterioration under multiple paraphrasing and when applied to shorter texts. We introduce *Pattern Stability Score (PSS)*, a novel detection framework that leverages local statistical features and stability dynamics across paraphrased variants. Specifically, the proposed method combines global and local z-score features with higher-order statistics of run-length patterns, enriched by autocorrelation signals and stability scores computed over paraphrase depth. Numerical evaluations are performed on PG-19, a large-scale long-form benchmark while systematically stress-testing robustness under up to eight rounds of paraphrasing with Mistral-7B. Compared to prior z-score thresholding baselines, our approach improves detection AUC (area under the receiver operating characteristic curve) by over 10–15 percentage points across different token lengths. Additionally, it achieves strong precision–recall balance and AUC greater than 0.95 at full length, demonstrating resilience where prior detectors collapse. Finally, sensitivity analysis is conducted on window size, stride, and token length to validate design choices. Overall, these empirical results establish PSS as a practical and extensible framework for watermark detection, highlighting stability-based features as a promising direction for safeguarding LLM outputs against potential adversarial paraphrasing.

## 1 Introduction

Large language models (LLMs) are now deployed at scale across both consumer and enterprise applications. As they increasingly integrate into writing workflows, the need to identify machine-generated content has shifted from a primarily academic inquiry to a practical requirement across domains such as education (Susnjak & McIntosh, 2024; Cotton et al., 2024), journalism (Chen & Shu, 2024; Zhou et al., 2023a), and science policy (Blau et al., 2024; Gao et al., 2023), among others. A long history of *post-hoc* detection has been explored. For example, GLTR (Gehrmann et al., 2019) leverages rank histograms from a reference LM to highlight text that disproportionately employs high-probability tokens. This approach is efficient but its reliance on rank features renders it vulnerable to paraphrasing and domain variation. Similarly, Grover (Zellers et al., 2019) jointly trains a generator–discriminator pair, using an in-domain classifier for detection. However, the performance declines when either the generator or domain changes, with paraphrasing further diminishing robustness. More recent work, such as Binoculars (Hans et al., 2024), compares likelihoods under two open LMs and applies a likelihood-ratio style criterion for zero-shot detection. This improves cross-domain generalization but still exhibits sensitivity to paraphrasing and short inputs. Other methods, including curvature- and rank-based tests such as DetectGPT and its variants (Mitchell et al., 2023), similarly rely on probability access from one or more LMs and remain susceptible to paraphrase smoothing.

In contrast, our focus is on *watermarking*, which offers several advantages: it is straightforward to deploy (via a lightweight, keyed bias during generation), requires only token identities on the detector side (removing dependence on proprietary probability distributions), and enables principled statistical testing under distribution shift due to its keyed structure. At a high level, watermarking operates as follows: the generator biases token selection toward a hidden "greenlist" so that downstream text exhibits detectable statistical structure, while remaining human-readable (Kirchenbauer

et al., 2023; Qu et al., 2025; He et al., 2025a; Lau et al., 2024). The standard detector aggregates evidence into a *global* z-score and compares it to a threshold. However, a determined adversary can paraphrase the text, diluting or locally rearranging this signal (Sadasivan et al., 2025; Cheng et al., 2025; Mitchell et al., 2023; Zhou et al., 2023b). There exist several methods to modify the watermarking scheme to make it more robust with respect to certain adversarial attacks such as paraphrasing (see Section 2.2 for a review on other types of watermarking schemes). These studies motivate our design choice: instead of changing the generator to resist paraphrasing, we change the detector to exploit signals that paraphrasing preserves only imperfectly—namely local structure and stability across rewrites. This detector-centric approach offers several critical advantages over modifying watermarking schemes. First, *deployment simplicity and compatibility* is paramount: simple watermarking schemes like greenlist watermarking have already been deployed at scale (Dathathri et al., 2024), with established infrastructure and known performance characteristics. Modifying detection algorithms requires no changes to existing generation pipelines, model architectures, or serving infrastructure, enabling immediate deployment across millions of already-watermarked texts. In contrast, advanced watermarking schemes often require architectural modifications, additional model components, or complex sampling procedures that increase latency and computational cost. Second, *retroactive applicability* provides immediate value: enhanced detectors can identify watermarks in previously generated content without requiring regeneration. Given the vast corpus of watermarked text already in circulation, improving detection provides immediate benefits, while new watermarking schemes only apply to future generations. This retroactive capability is particularly valuable for content verification in legal, educational, and journalistic contexts where historical text attribution is critical. Third, we achieve *robustness through defense-in-depth*: while sophisticated watermarking schemes like semantic embedding (Ren et al., 2024) or neural watermarking (Yu et al., 2025) show promise, they introduce new attack surfaces and failure modes. As noted in Diaa et al. (2025), increasing watermark complexity does not guarantee robustness—attackers can learn to exploit scheme-specific patterns. Our approach treats the watermark as a fixed signal and focuses on extracting maximum information through multiple statistical lenses (local, global, stability-based), providing defense-in-depth without relying on scheme secrecy. Fourth, we preserve *computational efficiency and generation quality*: advanced watermarking schemes often trade generation quality for robustness, requiring stronger biases or more aggressive vocabulary manipulation to survive paraphrasing. Simple schemes with improved detection preserve the delicate balance between watermark strength and text quality already optimized in production systems. Moreover, complex watermarking increases inference cost—multi-bit schemes (Xu et al., 2025; Feng et al., 2025) require multiple forward passes or auxiliary models, while our detection improvements add negligible overhead to the lightweight detection process. Finally, we approach *theoretical optimality within existing constraints*: recent theoretical work (Li et al., 2025a; He et al., 2025b) establish that for fixed watermarking schemes, there exist mathematically optimal detection rules. Rather than changing the watermark to chase robustness, we approach the theoretical detection limits for existing schemes through better statistical analysis. This aligns with the principle that detection should extract all available information before declaring the need for stronger watermarks.

Recent works on watermarking repeatedly highlight two open gaps: robustness to *multi-step paraphrasing* and stability on *short texts* (Sadasivan et al., 2025; Cheng et al., 2025). To address these gaps, we study a black-box adversary who can paraphrase any given text up to $K$ steps using a strong instruction-tuned LLM (here Mistral-7B-Instruct). The adversary does *not* know the watermark key or parameters and will preserve the original semantics and approximate length. Let $x^{(0)}$ denote the original passage (possibly watermarked) and $x^{(k)}{}_{k=1}^{K}$ its paraphrases at depths $D1 \ldots DK$. The detector receives a single text at test time (any $x^{(k)}$ for $k = 0, 1, \ldots, K$). Our objective is to maintain detection power under paraphrasing and across different text lengths while controlling false positives on human-written content.

**Why global z-score fails under paraphrasing.** Greenlist watermarking (Kirchenbauer et al., 2023) induces a binary indicator sequence over tokens (green/non-green) and a corresponding z-score measuring deviation from the null. The binary indicator sequence is constructed as follows: for each token $s^{(t)}$ in the generated text, we assign a value of 1 if the token belongs to the green list $G^{(t)}$ and 0 if it belongs to the red list $R^{(t)}$. Specifically, at each position $t$, a hash function seeded by the previous token $s^{(t-1)}$ deterministically partitions the vocabulary into a green list of size $\gamma|V|$ and a red list of size $(1-\gamma)|V|$, where $\gamma$ is typically 0.25 or 0.5. During watermarked generation, tokens from the green list are softly promoted by adding a bias $\delta$ to their logits. At detection time, we reconstruct this binary sequence by checking whether each observed token $s^{(t)}$ falls in its corresponding green

list $G^{(t)}$ (assigned 1) or red list $R^{(t)}$ (assigned 0), using the same hash function and seed. Global tests summarize all tokens into one statistic (e.g. z-score). Paraphrasing can break up long runs of green tokens, relocate them, or introduce local pockets of off-green context. These operations reduce the global statistic even when local evidence remains strong. The failure is structural: a single aggregate discards *where* evidence concentrates and *how* it behaves under edits.

**Key idea: stability-aware local detection.** The proposed method *Pattern Stability Score (PSS)*, is a detection framework that (i) extracts *local* watermark evidence via a rolling window and (ii) quantifies *stability* of that evidence across paraphrase depth. Specifically, we slide a window along the 0-1 sequence and compute several local statistics within each window. Note that when a window splits a consecutive sequence/run of ones, we expand it minimally so runs are not fragmented while shrinking the tail window to cover all remaining tokens. For each window we compute a 20-dimensional feature set: six

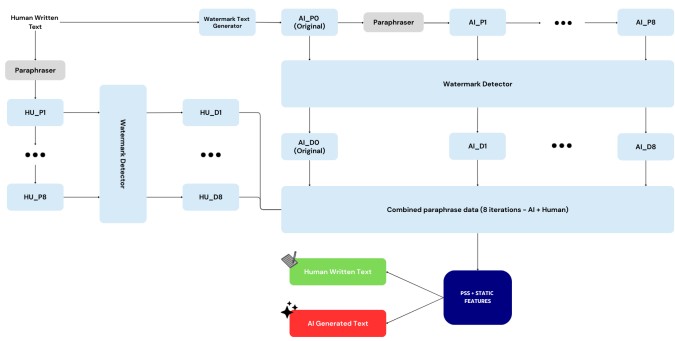

Figure 1: **End-to-end pipeline.** Human-written passages are paraphrased up to eight rounds (HU_P1 – HU_P8). A watermark text generator produces AI watermarked text AI_P0, then we paraphrased for AI and Human for 8 iterations to get all the P-files. After getting the detector data D-files from all the corresponding P-files (D-files of AI + Human), detection methods are applied.

summary statistics of the z-score sequence across windows (mean, variance, min, max, skew, and kurtosis), lag-1 and lag-2 autocorrelations of z-scores, six summary statistics of longest-run length in the binary sequence as well as the same six summary statistics of frequency of the longest run. Aggregating these per-window features yields robust *local* statistics. We then compute a *pattern stability* functional—PSS—over the trajectory $x^{(j)} \to \cdots \to x^{(K)}$, where Dj is the given text for some $j = 0, 1, \ldots, K - 1$. PSS is computed by extracting per-window local z-scores across all participating paraphrased versions ($Dj$ to $DK$), aligning them to the minimum window count for consistency, and computing the standard deviation across depths. Specifically, for each window position $w_i$, we calculate $\text{PSS}_i = \text{std}(z_i^{(j)}, \ldots, z_i^{(K)})$ where $z_i^{(k)}$ denotes the local z-score at window $i$ for depth $k$ for $k = 0, 1, \ldots, K$ while std(.) denotes the standard deviation function. This window-wise variability signal is then concatenated with 20 static features to form the complete feature vector (see Section 3.4 for more details). Then, a simple classifier (e.g., XGBoost or logistic regression) on these hybrid features produces the final decision using a 70/30 stratified train/test split. The rationale behind the proposed detector is that to improve the detection power among potential multi-step paraphrasing, the method fuses two main ingredients: (i) *local* rolling-window statistics that preserve spatial structure of watermark evidence (moments of local z-score, short-range autocorrelations, longest-run length and frequency), and (ii) uncertainty metric (PSS) that aggregates these features across paraphrase depths to capture both central tendency and variability (depth-wise variance and optional concordance). Local features expose pockets of concentrated green evidence that global tests average away, while PSS down-weights brittle depth-specific artifacts and rewards signals that persist under paraphrasing. This combination converts paraphrase-invariant regularities into separable features, yielding rather stable AUC (area under the receiver operating characteristic curve) at high depths and short lengths (see more details in Section 4). Figure 1 visualizes the end-to-end pipeline of the proposed methodology while Algorithm 1 in the Appendix provides the pseudocode.

The evaluation is conducted on a balanced corpus constructed from the PG-19 long-form book dataset (Rae et al., 2020). Specifically, we randomly sample 1,000 human-authored books and generate an equal number of watermarked passages using the greenlist watermarking method introduced in Kirchenbauer et al. (2023) applied with LLaMA-2-7B under configuration parameters $\gamma$=0.25 (greenlist ratio), $\delta$=1.5 (bias), and a fixed hash key. Texts are sampled via nucleus sampling with moderate temperature, with full details provided in Section 4. For consistency across conditions, all passages are truncated to 1,500 words, with additional experimental regimes established at 1,000, 500, and 300 words. To assess robustness, each passage is paraphrased for up to $K$=8 iterations

(depths $D1$–$D8$) using Mistral-7B-Instruct with a length-preserving prompt and fixed decoding hyperparameters. Performance is quantified using AUC, and complemented with sensitivity analyses over window size and stride. Our approach substantially narrows the robustness gap under both paraphrasing and reduced text lengths. For instance, at 1,500 words and depth $D6$, the proposed detector attains 94.6% AUC, markedly outperforming global z-score thresholding at 66.75%. Even at 300 words and $D6$, it preserves strong AUC of 81.27%, compared to 51.25% for the baseline. These numerical results underscore the effectiveness of the proposed framework and highlight its promise as a reliable watermark detection method.

Summary of main contributions are as follows:

- **Stability-driven detection.** We introduce the PSS, a principled measure that captures the persistence of watermarking signals across successive paraphrasing depths. Beyond formalizing this stability perspective, we demonstrate how PSS can be effectively integrated with *local* rolling-window statistics to enhance detection granularity.

- **Compact hybrid feature design.** We construct a 20-dimensional window-based feature set that incorporates statistical moments, autocorrelation descriptors, and run-length structural properties. This compact representation is deliberately engineered to maintain discriminative power even under aggressive paraphrasing and in short-text regimes, addressing key limitations of prior approaches.

- **Robustness under adversarial stress.** Through systematic evaluation on the PG-19 benchmark—subjected to up to eight rounds of paraphrasing and reduced passage lengths as short as 300 words—we show that PSS consistently surpasses global z-score baselines in AUC.

- **Comprehensive sensitivity analysis.** We analyze the influence of critical hyperparameters, including window size, stride, and input length, on detection performance. The empirical results confirm that the proposed method remains robust under moderate parameter variations, reinforcing the reliability and practical deployability of PSS in diverse settings.

The rest of the paper is organized as follows. Section 2 reviews watermarking and detection methods while Section 3 formalizes proposed methods, namely local features and PSS. Section 4 details datasets, paraphrasing, metrics, and then presents empirical results. Finally, Section 5 covers some concluding remarks, limitations, and future research directions. The Appendix contains extended numerical analyses, sensitivity test details, provided pseudocode, and details of LLM usage.

## 2 Related Work

### 2.1 Watermarking for LLMs

Greenlist watermarking biases token sampling toward a partition of the vocabulary determined by a keyed hash. Detection then tests whether the realized proportion of "green" tokens is unusually high under the null (Kirchenbauer et al., 2023). The standard detector reduces the problem to a single global z-score with a fixed threshold (often z-score $> 4$). Follow-up work characterizes trade-offs among bias strength, quality, and false positives, and analyzes limits under channel constraints and adversarial distortion (Qu et al., 2025; He et al., 2025a; Lau et al., 2024). These approaches assume that compressing evidence into one statistic retains power; in practice, global aggregation is fragile when text is paraphrased or short.

A line of work investigates how paraphrasing and distribution shift erode detector power. Paraphrasing attacks—produced by instruction-tuned models or controlled editing—can disperse local green runs, alter token-level dependencies, and reduce the global statistic while preserving semantics (Cheng et al., 2025; Sadasivan et al., 2025). Beyond watermark-specific detectors, post-hoc detectors such as DetectGPT and its accelerations exploit curvature or log-likelihood perturbations to separate human and model text, but they also degrade under paraphrases or domain shift (Mitchell et al., 2023; Zhou et al., 2023b). Our multiple rounds of paraphrasing follows this literature: a black-box paraphraser generates a depth-$K$ chain ($D1 \ldots DK$) without access to watermark keys, aiming to flip the detector while keeping meaning (Sadasivan et al., 2025; Cheng et al., 2025; Rastogi & Pruthi, 2024).

### 2.2 Adaptive and alternative watermarking schemes

Adaptive schemes modify partitioning or biasing as generation proceeds, or modulate the watermark via content- or entropy-aware policies (Feng et al., 2024; Lau et al., 2024). Theoretical analyses characterize fundamental limits, e.g., how much capacity is available for reliable marking under a given distortion budget and adversarial rewrite power (He et al., 2025a; Qu et al., 2025). Recent semantic approaches move beyond token-level manipulation, with SemaMark (Ren et al., 2024) in-

troducing semantic embeddings for vocabulary partitioning rather than token hashes, providing robustness to paraphrasing attacks by maintaining semantic consistency. Similarly, semantic invariant watermarks (Liu et al., 2024a) generate watermark logits based on semantic context using embedding models, while SAEMark (Yu et al., 2025) employs Sparse Autoencoders to embed watermarks through feature-based rejection sampling on neural activations. Production-scale deployment has been achieved with SynthID-Text (Dathathri et al., 2024), which introduces tournament sampling with provable non-distortion properties and serves over 20 million responses in Google Gemini, validating the feasibility of pattern-based approaches at scale. Publicly-detectable watermarking (Fairoze et al., 2025) achieves distortion-free watermarking with cryptographic signatures via rejection sampling, incorporating error-correction for low-entropy periods. These studies motivate our design choice: instead of changing the *generator* to resist paraphrasing, we change the *detector* to exploit signals that paraphrasing preserves only imperfectly—namely local structure and stability across rewrites.

**Local vs. global statistics for detection** Global tests ignore *where* evidence concentrates. Local analyses (rolling windows, run-length distributions, short-range autocorrelations) preserve spatial structure that is costlier for paraphrasers to randomize without semantic drift. Recent frequency-based approaches like FreqMark (Xu et al., 2024) employ Short-Time Fourier Transform for sentence-level detection with periodic signal embedding, achieving AUC up to 0.98 through windowing approaches that parallel our local detection strategy. Adaptive watermarking (Liu & Bu, 2024) uses entropy-based token selection with semantic logits scaling, selectively watermarking high-entropy distributions for improved robustness. Statistical frameworks (Li et al., 2025a) provide closed-form expressions for asymptotic error rates and mathematically optimal detection rules, while likelihood-based detection (Li et al., 2025b) estimates null token probabilities for accurate detection, achieving approximately 65% power improvement over baselines. Universal optimality results (He et al., 2025b) characterize minimum Type-II error for any watermarking scheme, establishing fundamental limits. Multi-bit approaches like MajorMark (Xu et al., 2025) implement clustering-based majority voting with block partitioning, while BiMark Feng et al. (2025) achieves 30% higher extraction rates for short texts through multilayer architecture with bit-flip unbiased mechanisms. Ensemble watermarks (Niess & Kern, 2025) combine acrostic patterns, sensorimotor norms, and red-green watermarks, achieving satisfactory detection rate compared to red-green alone after paraphrasing. Linguistic-feature or style-based detectors (Gehrmann et al., 2019; Wang et al., 2018; Gao et al., 2024) implicitly leverage locality but are unkeyed and risk false positives on atypical human styles. Our method remains keyed to the watermark while augmenting the global test with compact local statistics. Empirically, this hybrid design—local moments and autocorrelations of z-score, longest-run and its frequency—closes much of the robustness gap under paraphrasing and short lengths, while keeping computation modest and features interpretable.

### 2.3 Paraphrasing detection and inversion

Orthogonal to watermarking, paraphrasing-detection methods attempt to identify machine paraphrase patterns directly, e.g., by modeling machine paraphrasing behavior or by inverting paraphrases (Krishna et al., 2023; Wang et al., 2024). adaptive attacks using Direct Preference Optimization achieve over 96% evasion rate against surveyed watermarks (Diaa et al., 2025), while cross-lingual attacks (He et al., 2024) reveal fundamental weaknesses, with Cross-lingual Watermark Removal Attack decreasing AUCs from 0.95 to 0.67. Comprehensive evaluations (Liu et al., 2024b) show KGW achieving only 0.0349 watermark rate under paraphrase attacks, demonstrating the need for multi-attack robustness testing. Domain-specific challenges further complicate detection: SWEET (Selective WatErmarking via Entropy Thresholding) (Lee et al., 2024) addresses code's low entropy by watermarking only high-entropy segments, while medical text evaluation (Hastuti et al., 2025) shows current watermarking methods compromise medical factuality, introducing Factuality-Weighted Score metrics that prioritize accuracy over detectability. These approaches can complement watermark detectors but require assumptions about the paraphrasing model and are vulnerable when attackers switch paraphrasers. Our setting treats the paraphraser as a black box and remains agnostic to the specific model family, focusing instead on the behavior of watermark evidence under paraphrasing.

## 3 Proposed Methodology

In this section, we describe the proposed watermarking detector which includes the *local* statistics and the *Pattern Stability Score* (PSS) computed across paraphrase depth, accompanied with multiple paraphrasing, and the final classifier.

### 3.1 Preliminaries: greenlist watermarking and the global test (Kirchenbauer et al., 2023)

Let $\mathcal{V}$ be the vocabulary and let $h(\cdot; k)$ be a keyed hash that maps a token-context pair to $[0, 1]$. For a partition parameter $\gamma \in (0, 1)$, the *greenlist* at position $t$ is

$$G_t = \{v \in \mathcal{V} : h(v, x_{<t}; k) \leq \gamma\},$$

where $x_{<t} = (x_1, x_2, \ldots, x_{t-1})$ denotes the sequence of tokens preceding position $t$. During generation, the model increases the probability mass on $G_t$ by a bias $\delta > 0$. Given a token sequence $x_{1:n}$, define the indicator $b_t = \mathbf{1}\{x_t \in G_t\}$ and the global test statistic (i.e. z-score)

$$z(x_{1:n}) = \frac{\sum_{t=1}^n b_t - n\gamma}{\sqrt{n\gamma(1-\gamma)}}.$$

Classical detection declares "watermarked" if z-score $> \tau$ for a fixed threshold (often $\tau{=}4$). This test is efficient and interpretable but discards spatial information and is known to degrade under paraphrasing (Sadasivan et al., 2025).

### 3.2 Multi-step paraphrasing and data generation

We assume a black-box paraphraser that maps any text $x^{(0)}$ to a sequence of paraphrases $\{x^{(k)}\}_{k=1}^K$, preserving semantics and approximate length, without access to $k$ or $(\gamma, \delta)$. In our pipeline: (i) watermarked texts are generated with an open LLM using standard $(\gamma, \delta)$ and sampling; (ii) each text is paraphrased up to depth $K{=}8$; (iii) detection is run on any single depth at test time. Exact prompts and decoding settings are specified in Section 4. We evaluate multiple lengths, i.e. $n \in \{300, 500, 1000, 1500\}$ tokens.

### 3.3 Local rolling-window statistics

Global aggregation ignores *where* watermark evidence concentrates. We therefore compute *local* features by sliding a window of size $w$ with stride $s$ across $x_{1:n}$ and its indicator sequence $b_{1:n}$ to compute local z-scores[1]. Let the $i$-th window cover indices $t \in [a_i, b_i]$. For each window we compute:

1. **Local z-score summary** over $\{b_t\}_{t=a_i}^{b_i}$ defined as $z_i = \frac{\sum_{t=a_i}^{b_i} b_t - m_i\gamma}{\sqrt{m_i\gamma(1-\gamma)}}$, $\qquad m_i = b_i - a_i + 1$,

   and compute six summary statistics of $\{z_i\}$ sequence across windows: mean, variance, min, max, skew, kurtosis.
2. **Autocorrelation of local z-score** across windows: $\rho_z(\ell)$ at lags $\ell \in \{1, 2\}$.
3. **Run-length statistics** inside the window: the longest consecutive run of ones $R_i$, and its frequency $F_i$ (number of occurrences). We then compute the six summary statistics of $\{R_i\}$ and of $\{F_i\}$ across windows.

This yields a compact 20-dimensional *local feature set*: 8 from z-score (6 summary statistics + 2 autocorrelations), 6 from run-length, and 6 from run-frequency. These capture spatial concentration and short-range dependencies that paraphrasers disrupt only imperfectly without semantic drift.

### 3.4 PSS across paraphrase depth

Paraphrasing aims to rearrange evidence. If the underlying text is watermarked, we expect local evidence to persist across mild rewrites; for human text, local evidence should fluctuate around the null. We formalize this intuition via a stability functional over the local z-score trajectories $\{x^{(k)}\}_{k=0}^K$. across paraphrase depths. Given a text at an unknown paraphrase depth $j \in 0, 1, \ldots, K$, we generate its subsequent paraphrases up to depth $K$ to obtain the sequence $\{x^{(k)}\}_{k=j}^K$. For each text in this sequence, we compute local z-scores using the rolling-window procedure described in Section 3.3. Because different paraphrase depths can yield different numbers of windows, we align all sequences to the minimum window count across depths before aggregation. The Pattern Stability Score is then computed by measuring the variability of local z-scores at each window position across depths. Specifically, for each aligned window position $i$, we calculate:

$$\text{PSS}_i = \text{std}(z_i^{(j)}, z_i^{(j+1)}, \ldots, z_i^{(K)}),$$

where $z_i^{(k)}$ denotes the local z-score at window position $i$ for paraphrase depth $k$. This computation yields a vector of stability scores, one for each window position, capturing how consistently the

---

[1] We use $w{=}50$, $s{=}10$ by default. If a window boundary splits a consecutive run of ones in $b_{1:n}$, we expand the window minimally to keep the run intact; the tail window shrinks to cover remaining tokens. Sensitivity to $(w, s)$ is reported in the Appendix.

watermark signal manifests at each local region across paraphrasing transformations. The intuition behind PSS is that watermarked text exhibits more stable local patterns across paraphrases compared to human text. When a text is genuinely watermarked, the underlying statistical bias persists even as surface tokens change through paraphrasing, resulting in relatively consistent local z-scores and thus lower PSS values at each window position. Conversely, human text subjected to paraphrasing shows higher variability in local z-scores across depths, as there is no underlying watermark signal to maintain consistency. The complete feature vector for classification consists of two components: (i) the PSS values computed across all window positions, providing a stability profile of the text, and (ii) the static features extracted from the current text, including summary statistics of z-scores, run-length patterns, and frequency statistics as defined in Section 3.3. This hybrid approach combines the temporal stability information from PSS with the instantaneous statistical patterns from static features.

### 3.5 Classifier and decision rule

Given a passage at an unknown paraphrase depth $Dj$, we (i) compute the greenlist indicator $b_{1:n}$ and local z-scores, (ii) extract the *local* rolling-window feature set from Section 3.3 (20-dimensional statistical values, short-range autocorrelations (lags 1 and 2), and run-length statistics), and (iii) optionally augment these with *stability* features via the PSS from Section 3.4, which aggregates depth-wise consistency of the same local statistics. The resulting feature vector $g(x)$ is fed to a lightweight supervised classifier that outputs a posterior $p_\theta(y{=}1 \mid g(x))$ and a binary decision via a fixed threshold. We compare four standard learners on $g(x)$—logistic regression, random forest, XGBoost, SVM (RBF), and $k$NN—chosen for complementary bias/variance profiles and interpretability. Classifier hyperparameters use library defaults unless noted. Unless stated, XGBoost serves as the representative classifier in numerical results (it consistently ranks first or tied in our empirical studies). For each depth and input-length regime, we use a stratified 70/30 train/test split with fixed random seed. Hyperparameters follow library defaults unless noted, no test-time tuning is performed. At inference, we report AUC, together with Precision/Recall/F1. Figure 1 summarizes the modules and data flow.

## 4 Experiments and Results

We evaluate robustness to paraphrasing and short texts on a balanced long-form corpus. This section specifies the data and multiple paraphrasing, the compared methods, the training/evaluation protocol, and the main numerical results including some robustness. Extended tables and plots are deferred to the Appendix while the code and data scripts are available at the following anonymous link: `github.com/mastercoder0368/PSS-Watermark-Detection`.

### 4.1 Experimental setup

We sample randomly selected 1,000 human written books from PG-19 and generate 1,000 watermarked passages for parity. Each passage is truncated to a fixed token length. We report four length regimes: 300, 500, 1,000, and 1,500 tokens. We use standard greenlist watermarking with partition parameter $\gamma{=}0.25$, bias $\delta{=}1.5$, and a fixed hash key. Decoding uses nucleus sampling and moderate temperature, the detector uses the same $(\gamma, \delta)$ and hashing as the generator. Finally, an instruction-tuned paraphraser (Mistral-7B-Instruct) rewrites each passage up to depth $K{=}8$ under a length-preserving prompt. We denote the original as $D0$ and the $k$-th paraphrase as $Dk$. At test time the detector receives *one* text (potentially any $Dk$) without access to other depths.

### 4.2 Methods compared

We compare four detectors that differ only in feature design, all use the same training protocol.

1. **Global z-score threshold:** The canonical one-sided test declares "AI" if z-score $\geq \tau$ with $\tau{=}4$, else "Human".

2. **Local z-scores (20-D):** We compute rolling-window local $z_i$ across the passage. We then form a *fixed-length vector* from the first 20 window $z_i$'s (if more windows exist, we uniformly subsample to 20; and we make sure the included sequence have at least 20 z-scores). This 20-dimensional (20-D) *raw local pattern* is fed to LR/RF/XGB/SVM/KNN classifier models.

3. **Static features (20-D).** Instead of raw $z_i$ values, we summarize *across windows* using compact statistics that preserve locality signals while reducing dimensionality: for $\{z_i\}$, six moments (mean/var/min/max/skew/kurtosis) plus lag-1 and lag-2 autocorrelations (8 features), for the longest run length per window $\{R_i\}$, six moments (6 features); and for the frequency of the longest run per window $\{F_i\}$, six moments (6 features). In total, there are 20 features. These features are provided to the same classifier models.

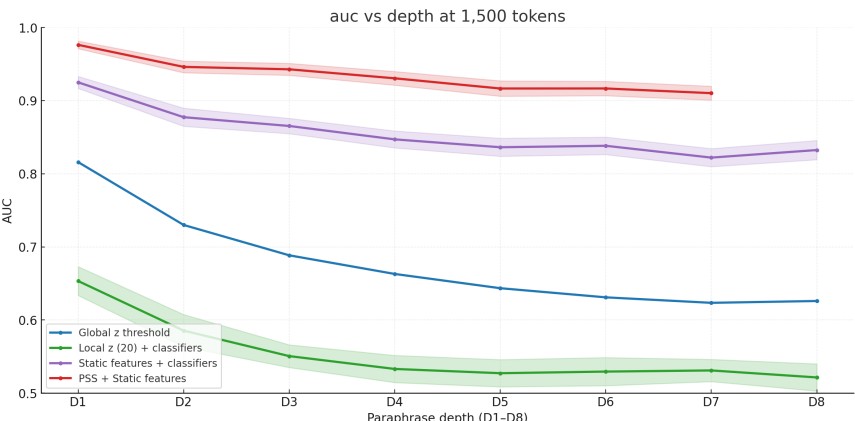

Figure 2: **AUC vs. paraphrase depth at 1,500 tokens.** Mean AUC (solid lines) with $\pm 1$ SD bands (shaded) over 30 runs with random 70/30 splits for four methods—Global z threshold, Local z (20) + classifiers, Static features + classifiers, and PSS + Static—all using XGBoost. Local statistics improve robustness relative to the global threshold, and adding PSS further flattens the AUC decline from $D3$–$D8$.

4. **PSS + static features.** We compute PSS values as described in Section 3.4 over paraphrase depth using the same local statistics to quantify depth-wise consistency (low dispersion, concordant trends). Then, we concatenate PSS with a selected subset of the static features and train an XGBoost classifier (our best single model).

### 4.3 Empirical results

**Performance at 1,500 tokens.** Figure 2 depicts AUC against paraphrase depth $D1$–$D8$. The global z-score threshold is competitive at shallow depth but declines steadily as paraphrasing deepens. Injecting locality slows this drop: both *local z (20-D) + classifiers* and static features + classifiers yield flatter AUC curves than the global statistic. The largest gains come from adding *stability—PSS + Static* remains comparatively flat through mid/late depths, indicating that cross-depth persistence provides signal beyond any single local snapshot. Noe that results for PSS + Static are provided up to depth 7 only since based on its definition, the proposed PSS + Static method requires at least one additional paraphrased text.

**Shorter texts.** Figure 3 shows AUC vs. depth for 1,000/500/300 tokens (top-left, top-right, bottom-left) and AUC vs. token length at $D7$ (bottom-right). As sequences shorten, all methods degrade, reflecting reduced evidence. Nevertheless, locality and stability remain beneficial: *static features* consistently outperform global baselines across depths, and *PSS + static* retains the largest margins, particularly beyond $D3$ demonstrating resilience when text is short and paraphrasing is deep. At $D7$, our method is most accurate across all lengths, with the gap widening around 500–1,000 tokens.

**Paraphraser independence.** We further test an alternating *mix* schedule—Mistral-7B-Instruct at $D1$, Qwen2-7B-Instruct at $D2$, then alternating through $D8$. Figure 4 shows *PSS + static* maintains the leading curve and degrades more slowly than alternatives, mirroring the single-paraphraser case. This indicates that the stability cue captured by PSS is not tied to idiosyncrasies of a particular paraphraser. For completeness, we also evaluate single-model paraphrasers (Gemma-7B-IT and Qwen2-7B-Instruct); rankings and margins remain consistent—see Appendix A.1 for full tables.

In summary, the satisfactory numerical performance of the proposed method is that paraphrases often *redistribute*

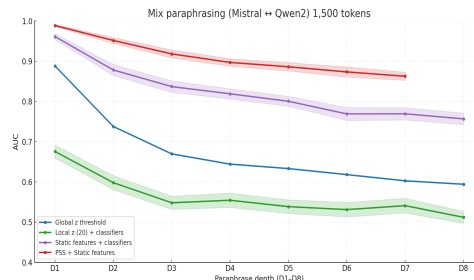

Figure 4: **Mix paraphrasing (Mistral $\leftrightarrow$ Qwen), 1,500 tokens.** AUC vs. depth under alternating paraphrasers.

watermark evidence rather than eliminate it. A single global statistic can be deflated by fragmenting long green runs, but doing so *consistently across windows and across depths* is harder without semantic drift. Local moments and short-range autocorrelations recover pockets of concen-

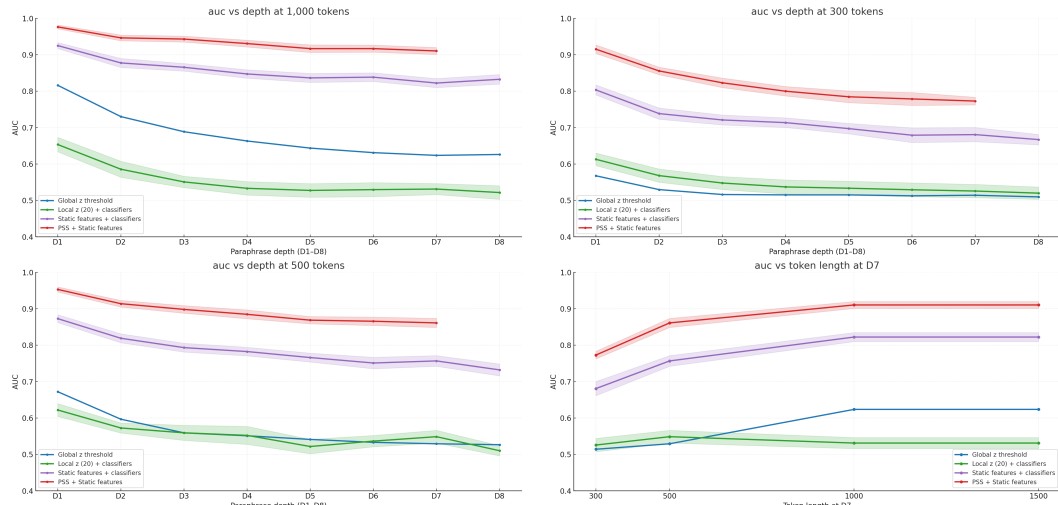

Figure 3: **Shorter texts.** Mean AUC (solid) with $\pm 1$ SD bands (shaded) over 30 random 70/30 splits. Top-left to bottom-left: AUC vs. paraphrase depth for 1,000/500/300 tokens; bottom-right: AUC vs. token length at $D7$. All methods degrade with less text, but local/static features mitigate the drop and PSS + Static maintains the strongest performance across depths and lengths, including at $D7$.

tration; run-length features react to fragmentation, and PSS converts depth-wise persistence (low dispersion and concordant trends across $Dk$) into a compact signal.

## 5 Concluding Remarks

We presented a stability-aware detector for watermarked LLM text that fuses *local* rolling-window statistics with a specific stability score (i.e. PSS) computed across paraphrase depth. By preserving spatial structure—moments and short-range autocorrelations of local z-scores, run-length and run-frequency features—and then summarizing consistency over $D0 \rightarrow D8$, the method outperforms global z-score thresholding and its classifier variants, with graceful degradation down to 300 tokens. The detector is keyed, simple to implement, and incurs low inference overhead, making it practical for real-world attribution settings.

**Limitations.** Our multi-step paraphrasing threat model assumes a black-box paraphraser without access to the watermark key, thereby preserving the confidentiality of watermarking. While sharing the watermark key could enhance detection performance, it would compromise key secrecy. Moreover, we do not evaluate *adaptive* attackers that are explicitly trained to minimize PSS. Our empirical analysis is further constrained to English long-form prose, excluding domains such as source code, poetry, or highly technical writing. Experiments are limited to a single human-authored corpus (PG-19) and one watermark configuration; thus, broader datasets, languages, and key variations remain to be investigated. Finally, supervised calibration may be sensitive to distribution shift, and the robustness of thresholds and scores across domains has yet to be systematically established.

**Future work.** Future work will extend both the methodological scope and the evaluation framework. We intend to examine stronger adversarial settings, including human-authored paraphrases, cross-lingual rewritings, and paraphrasing models explicitly trained to obscure stability signals. Another line of investigation is adapting the detector to handle mixed-authorship and partially watermarked documents, with the goal of localizing short watermarked segments. A particularly promising direction lies in joint generator–detector design, where watermark policies are co-tuned to preserve the local structure leveraged by PSS while systematically exploring the trade-offs among capacity, utility, and stability. We also aim to advance generalization and calibration across diverse corpora and languages by integrating uncertainty quantification and distribution-drift monitoring to support long-term deployment. Finally, we plan to pursue efficiency improvements, including compressing feature representations, designing lower-cost surrogates for PSS, and benchmarking throughput on commodity hardware.

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

# A    Appendix

In this appendix, we provide complementary material to support the main text. Section A.1 reports sensitivity tests, including robustness to different paraphrasers (Gemma-7B-IT and Qwen2-7B-Instruct) and a window/stride sweep with a heatmap summary (Section A.2). Section A.3 details compute and implementation notes, and it also includes dataset-level pseudocode matching our implementation. Finally, Section A.4 contains the required LLM usage disclosure.

**Setup recap.** We evaluate four families of detectors: (1) *Global z threshold*; (2) *Local z (20)* (first 20 rolling-window local z's as features); (3) *Static features* (windowed moments, short-range auto-correlations, run-length and run-frequency summaries); and (4) *PSS + static* which augments static features with *Pattern Stability Scores* computed from standard deviation of aligned local z-score trajectories across depths. All models use XGBoost for classifier-based lines unless stated; windows use $w=50$, stride $s=10$, with the non-fragmenting expansion rule. This mirrors the main-text configuration to avoid confounds.

## A.1    Sensitivity to paraphraser choice

**Motivation.** Retroactive detectors sometimes latch onto paraphraser-specific artifacts. Our detector explicitly targets *cross-depth stability* of local watermark evidence, which should persist irrespective of the paraphrasing model. We therefore re-run the entire pipeline with two single-model paraphrasers beyond Mistral-7B-Instruct used in the main text: *Gemma-7B-IT* and *Qwen2-7B-Instruct*. For each, we generate $D1$–$D8$ chains under the same prompts and decoding settings as in Section 4.

**Results.** Across both paraphrasers, the ordering of methods is consistent with the main text: global thresholding drops fastest with depth; injecting locality slows degradation; and *PSS + static* yields the flattest curves and the highest accuracies at mid/late depths. In particular, the stability signal is additive to locality, preserving margins even when token distributions shift due to a different rewriting policy. Full AUC values per depth are reported in Tables 1 and 2.

Table 1: **Gemma-7B-IT paraphrasing: AUC (%) vs. depth ($D1$–$D8$).** All classifier entries use XGBoost; values are mean $\pm$ std over 30 runs.

| Method | D1 | D2 | D3 | D4 | D5 | D6 | D7 | D8 |
|---|---|---|---|---|---|---|---|---|
| Global z-score threshold | 56.50 ± 0.00 | 54.35 ± 0.00 | 53.75 ± 0.00 | 53.55 ± 0.00 | 53.25 ± 0.00 | 53.30 ± 0.00 | 53.05 ± 0.00 | 53.00 ± 0.00 |
| Local z-score (20) | 66.07 ± 1.42 | 59.03 ± 1.66 | 58.92 ± 1.99 | 57.52 ± 2.20 | 57.81 ± 1.63 | 57.32 ± 1.40 | 57.55 ± 1.55 | 57.80 ± 1.41 |
| Static features | 72.75 ± 1.87 | 68.95 ± 1.87 | 67.02 ± 1.29 | 68.10 ± 1.35 | 67.48 ± 1.76 | 67.10 ± 1.83 | 66.82 ± 1.58 | 65.16 ± 1.87 |
| **PSS + Static** | **83.51 ± 1.43** | **77.85 ± 1.63** | **75.98 ± 1.45** | **76.54 ± 1.21** | **73.69 ± 1.49** | **73.33 ± 1.23** | **72.92 ± 2.03** | – |

Table 2: **Qwen2-7B-Instruct paraphrasing: AUC (%) vs. depth ($D1$–$D8$).** All classifier entries use XGBoost; values are mean $\pm$ std over 30 runs.

| Method | D1 | D2 | D3 | D4 | D5 | D6 | D7 | D8 |
|---|---|---|---|---|---|---|---|---|
| Global z-score threshold | 73.25 ± 0.00 | 69.90 ± 0.00 | 69.00 ± 0.00 | 68.40 ± 0.00 | 67.90 ± 0.00 | 67.70 ± 0.00 | 67.35 ± 0.00 | 66.80 ± 0.00 |
| Local z-score (20) | 58.90 ± 1.61 | 58.15 ± 1.69 | 55.03 ± 1.71 | 55.48 ± 1.64 | 55.86 ± 1.66 | 55.19 ± 1.64 | 54.27 ± 1.63 | 55.27 ± 1.66 |
| Static features | 82.08 ± 1.45 | 78.64 ± 1.55 | 77.15 ± 1.51 | 77.89 ± 1.48 | 77.47 ± 1.52 | 78.37 ± 1.50 | 78.77 ± 1.49 | 76.36 ± 1.47 |
| **PSS + Static** | **94.67 ± 0.68** | **92.68 ± 0.79** | **92.38 ± 0.82** | **92.63 ± 0.78** | **91.90 ± 0.91** | **90.57 ± 0.98** | **90.47 ± 1.02** | – |

**Takeaway.** Consistent rankings across paraphrasers support the claim that stability-aware local detection is *paraphraser-agnostic*. The detector exploits invariants (local concentration and cross-depth persistence) that are difficult to erase simultaneously without semantic drift or length distortion.

## A.2    Additional sensitivities

**Window and stride.** Figure 5 summarizes the *mean* AUC across depths ($D1$–$D8$) for all $(w, s) \in \{40, 50, 60\} \times \{5, 10, 15\}$ at 1,500 tokens. Performance is highly stable: the best setting $(50, 15)$ achieves 99.0%, while the lowest $(50, 10)$ records 95.95%, a spread of only 3.05 percentage points. The configuration $(50, 10)$ used throughout yields 95.95%, lying well within this plateau, confirming that our detector remains robust to moderate changes in window size and stride.

**Classifier choice.** Using LR/RF/XGB/SVM/$k$NN for local/static features yields the same ordering. XGBoost is typically best. Gains primarily trace to feature design rather than model complexity.

## A.3    Compute, implementation, and qualitative examples

**Hardware and runtime.** Experiments ran on A100-40GB GPUs with 64,GB host RAM. Paraphrasing/detection for a full depth chain ($D1$–$D8$) with 1,500 tokens per document typically took ~2 days per batch (one GPU per job). Detector-side inference is lightweight: computing green indicators, window features, and PSS is $O(n)$ in text length with memory linear in $n$. Specifically,

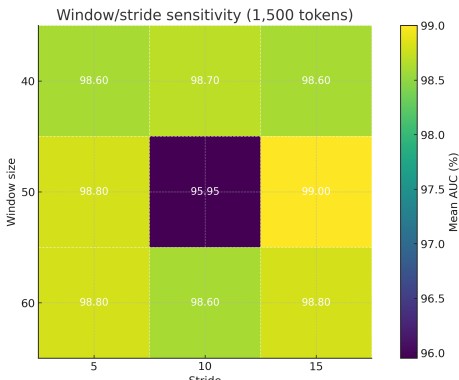

Figure 5: **Window/stride sensitivity (1,500 tokens)** for *PSS + static*. Numbers show mean AUC (%) across depths $D1$–$D8$.

preprocessing is $O(n)$ to compute $b_{1:n}$ and windows; feature aggregation is $O(n/w)$ for window size $w$ (defaults: $w{=}50$, stride 10). Classifier inference is $O(d)$ in the feature dimension ($d{=}20$ for local-only; slightly larger with PSS). Memory is linear in $n$. We implemented the proposed algorithm in Python with standard libraries.

**Implementation notes.** We implement watermarking and detection in Python. Local windows use the non-fragmenting rule that minimally expands a window to avoid cutting through consecutive green runs, shrinking the final tail window if needed to keep coverage. PSS aligns local z sequences across depths to the minimum window count before computing per-position standard deviation.

**Pseudocode: PSS + Static (dataset-level, matches implementation)**

---

**Algorithm 1** PSS + Static: training/evaluation from precomputed rolling-window CSVs

---

**Require:** CSVs for depths D1..D8, each with columns: `id`, `label`, `z_score_*` (one per window), and STATIC_FEATURES; a set of depth sequences (e.g., `D1--D8`, `D2--D8`, ...); split ratio (0.7/0.3), random seed, XGBoost hyperparameters.
1: **for each** experiment $\mathcal{E} = \{d_{\min}, \ldots, d_{\max}\}$ **do**
2:    Load data frames $\{\mathrm{DF}_d\}_{d \in \mathcal{E}}$.
3:    For each $d$, collect z-window columns $\mathcal{C}_d = \{c : c \text{ starts with } \mathtt{z\_score\_}\}$.
4:    $n_{\mathrm{win}} \leftarrow \min_{d \in \mathcal{E}} |\mathcal{C}_d|$   (align depths by truncating to the minimum #windows).
5:    Build $Z_d \in \mathbb{R}^{N \times n_{\mathrm{win}}}$ from the first $n_{\mathrm{win}}$ z-window columns of $\mathrm{DF}_d$ (same row order across depths).
6:    Stack $\{Z_d\}_{d \in \mathcal{E}}$ along a new axis to get $T \in \mathbb{R}^{N \times n_{\mathrm{win}} \times |\mathcal{E}|}$.
7:    **PSS (windowwise variability over depths):** $P \leftarrow \mathrm{std}(T \text{ along depth axis}) \in \mathbb{R}^{N \times n_{\mathrm{win}}}$.
8:    Let $\mathrm{DF}_{\mathrm{base}} \leftarrow \mathrm{DF}_{d_{\min}}$ (first depth of the experiment).
9:    Extract `meta` $\leftarrow$ `DF_base[{id, label}]`,  `static` $\leftarrow$ `DF_base[STATIC_FEATURES]`.
10:   Form `full_df` by concatenating `meta`, **PSS** (columns `pss_win1..pss_winn_win`), and `static`; impute missing values with 0.0.
11:   $X \leftarrow$ `full_df` without `id,label`;  $y \leftarrow$ `full_df['label']`.
12:   Stratified train/test split (test_size = 0.3, random_state = 42).
13:   Train XGBoost (binary logistic, `eval_metric=auc`, `n_estimators=600`, `max_depth=6`, `learning_rate=0.05`, `subsample=0.85`, `colsample_bytree=0.8`, `n_jobs=-1`, `random_state=42`).
14:   Predict labels and probabilities on the test set; compute ROC–AUC, Precision, Recall, F1, and confusion matrix.
15:   Append metrics to the results table.
16: **end for**
17: Save the aggregated results to CSV.

---

## A.4   LLM usage disclosure

We used a large language model solely to aid and/or polish our writing.

