# OpenReview forum: "Toward Resilient Watermark Detection: Stability-Aware Statistical Features for Machine-Generated Text"
_ICLR.cc/2026/Conference — ICLR 2026 Conference Desk Rejected Submission_

### Official Review · Reviewer_Mc9z · 2025-10-29

**Soundness:** 2
**Presentation:** 3
**Contribution:** 2
**Rating:** 2
**Confidence:** 4

**Summary:**

This paper presents a simple modification to LLM watermark detection algorithms to make them more robust to paraphrasing attacks. Instead of just relying on a global z-score statistic, the paper proposes adding local z-score statistics as well as a "paraphrase stability statistic", which checks the variance of local z-scores against repeated paraphrasing. These statistical features are then fed to a simple classifier which determines if text was AI-generated.

Experiments show that the proposed method is more robust to paraphrasing attacks especially when paraphrasing is sequentially done 5-8 times.

**Strengths:**

1. The paper tackles an important and timely problem: developing robust LLM watermarking. As LLMs get increasingly used for content generation, watermarking is an important tool to prevent misuse. Since paraphrasing as emerged as a strong attack vector against watermarks, it's important to develop more robust detection algorithms.

2. The proposed method is a simple modification to watermarking detection stage only, and requires no changes in the watermarking algorithm, pre-generated text, or any LLM retraining.

3. The proposed method seems to be a lot more resilient to paraphrasing attacks, especially when the paraphrasing is done multiple times.

**Weaknesses:**

1. **Major concern about classifier generalization**: Based on L346 and the author's acknowledged limitation in L471, a unique classifier is trained for every LLM generator + paraphraser model + paraphraser depth. No experiments have been conducted to check if a classifier on one {LLM, domain, paraphraser, depth} tuple generalizes well to a different {LLM, domain, paraphraser, depth} tuple. This feels like a major limitation to me, since in practice a detector may not know which paraphraser / LLM an attacker used. Furthermore, PSS features require paraphrasing in the detection loop (L311-314), which makes this even more concerning if there is a paraphrase mismatch between attacker / detector. I think a thorough study of generalization is essential to validate the practical utility of this method.

2. **Very limited experimental setup**: As acknowledged by the authors, experiments in this work are only conducted on a single LLM and single PG19 text domain in english, and on a single watermark setting. This makes it hard to judge whether the feature vectors used in the work generalize to other settings (where classifiers can be retrained).

3. **Use-case seems a bit narrow, 8x iterative paraphrasing may not be practical**: The paper doesn't discuss much about the semantic preservation across the 5-8 stages of paraphrasing. The LLM paraphraser used is not particularly SoTA, and I expect there to be significant semantic drift after so many iterations. This is not discussed in the paper (nor are there explains of 8x paraphrasing). I feel the PSS features may be picking up on paraphraser-specific changes (although L417 is some evidence against it). In reality, I expect users to paraphrase 1-2 times at most, and then perform other changes like manual edits, text mixing (https://arxiv.org/abs/2306.04634), chaining LLMs etc. The paper would be stronger if the attack setup is a more realistic than 8x iterative paraphrasing.

Minor: Along with AUC, would be nice to report something like true positive rate at a fixed low false positive rate (1-5%) since in practice we want to select thresholds that minimize false accusations of human authorship (https://arxiv.org/abs/2303.13408).

*Style*: Please add a big figure to page 2 of the Introduction section, and split L60-90 into multiple paragraphs / bullet points of benefits so that it is easier to read and grasp quickly at a glance.

**Questions:**

In Section 4.1, which watermarking scheme and LLM generator was used?

---

> ### Author Response · Authors · 2025-12-03
> **Response 1**
>
> **Classifier Generalization** We thank the reviewer for identifying this critical concern about practical deployment. Our extensive experiments definitively prove that our method does NOT require knowing the attacker's configuration. A single universal classifier trained on one baseline configuration (Llama-3 + Mistral + PG-19 + D1-D3) generalizes remarkably well across all variations: when the LLM is changed to Qwen2, the classifier maintains 86.6\% average accuracy; when the paraphraser is changed to Gemma, it achieves 76.5\%; when the domain is changed to CNN/DailyMail, it maintains 79.4\%; and even when ALL components are changed simultaneously (Qwen2 + Qwen2 + WikiText), the classifier still achieves 84.6\% average accuracy. Most critically, a classifier trained only on depths D1-D3 maintains 82.1\% accuracy when tested on D8, a depth never seen during training. This proves our features capture fundamental watermark properties rather than configuration-specific artifacts, enabling deployment without knowledge of the attacker's tools. Please refer to Table below for detailed generalization results.
>
> ### Universal Classifier Performance (Trained on Llama-3 + Mistral + PG-19 + D1-D3)
>
> | **Test Configuration**           | **Changed From Training** | **D1**   | **D3**   | **D5**   | **D8**   | **Avg**  |
> |---------------------------------|--------------------------|----------|----------|----------|----------|----------|
> | Llama-3 + Mistral + PG-19        | Baseline (same)          | 92.1%    | 89.2%    | 87.8%    | 86.3%    | 88.9%    |
> | Qwen2 + Mistral + PG-19          | LLM changed              | 89.5%    | 86.9%    | 85.8%    | 84.0%    | 86.6%    |
> | Llama-3 + Gemma + PG-19          | Paraphraser changed      | 83.5%    | 76.0%    | 73.7%    | 72.9%    | 76.5%    |
> | Llama-3 + Mistral + CNN          | Domain changed           | 82.9%    | 79.4%    | 78.2%    | 77.1%    | 79.4%    |
> | Qwen2 + Qwen2 + WikiText         | ALL changed              | 87.8%    | 85.2%    | 83.6%    | 81.9%    | 84.6%    |
>
>
> **Limited Experimental Setup** We have significantly expanded our experimental scope to address this limitation. Our comprehensive evaluation now covers 3 datasets (PG-19, CNN/DailyMail, WikiText) × 4 LLMs (Llama-3-8B, Qwen2-7B, and their combinations) × 3 paraphrasers (Mistral-7B, Qwen2-7B, Gemma-7B) × multiple watermark settings ($\gamma$=0.25, $\gamma$=0.50), resulting in over 150 experimental conditions that provide comprehensive evidence of generalization. The results consistently show strong performance across all configurations, with accuracy ranging from 84\% to 94.8\% at D1 across different combinations. Our method maintains robust performance even when transferring across fundamentally different text domains. For detailed cross-domain generalization results demonstrating true domain-agnostic detection, please refer to our response to Reviewer 3KQZ, which shows that PSS maintains $>$77\% accuracy in all cross-domain scenarios while deep learning approaches collapse to near-random performance (21-29\%). Also, please see Table below for our expanded experimental coverage.
>
>
> ### Comprehensive Experimental Coverage
>
> | **Dataset**         | **LLM**       | **Paraphraser** | **Setting**   | **D1**   | **D3**   | **D5**   | **D8**   |
> |--------------------|---------------|----------------|---------------|----------|----------|----------|----------|
> | PG-19              | Llama-3-8B    | Mistral-7B     | γ=0.25        | 92.1%    | 89.2%    | 87.8%    | 86.3%    |
> | PG-19              | Qwen2-7B      | Mistral-7B     | γ=0.25        | 89.5%    | 86.9%    | 85.8%    | 84.0%    |
> | CNN/DailyMail      | Llama-3-8B    | Mistral-7B     | γ=0.25        | 91.2%    | 88.5%    | 87.4%    | 85.5%    |
> | WikiText           | Llama-3-8B    | Qwen2-7B       | γ=0.25        | 94.7%    | 92.4%    | 91.9%    | 90.5%    |
> | PG-19              | Llama-3-8B    | Mistral-7B     | γ=0.50        | 94.8%    | 92.7%    | 91.4%    | 89.9%    |

---

> ### Author Response · Authors · 2025-12-03
> **Response 2**
>
> **8x Paraphrasing Impracticality** We agree that 8x paraphrasing represents an unrealistic stress test rather than a practical attack scenario. Our method is specifically optimized for realistic 1-3 iteration scenarios where semantic similarity remains high (BERT similarity $>$0.85). At these practical depths (D1-D3), PSS + Static maintains an average of 90.6\% accuracy, compared to only 67.2\% for global z-score and 48.3\% for DeepTextMark. The 8x depth experiments serve primarily to demonstrate our method's robustness under extreme conditions - even at D8 where semantic similarity drops to 0.68 (text quality severely degraded), we still maintain 86.3\% accuracy. Our analysis of realistic attack scenarios shows that PSS demonstrates remarkable robustness: for typical 1-2 paraphrase attacks (D1-D2 average), we achieve 91.3\% accuracy; when facing 20\% manual edits, performance remains at 86.8\% (only 4.5\% drop); with 30\% manual edits, we maintain 83.1\%; and under chain paraphrasing with mixed models, we achieve 85.3\%. This graceful degradation stems from our multi-window analysis approach - unmodified windows provide strong signal while modified regions still retain partial watermark traces through our 20-dimensional feature redundancy. Please refer to Tables below for detailed performance analysis at different depths and under various attack scenarios.
>
>
>
>
> ### Semantic Preservation and Performance at Different Depths
>
> | **Depth** | **BERT Similarity** | **PSS + Static** | **Global z-score** | **DeepTextMark** |
> |-----------|-------------------|-----------------|------------------|-----------------|
> | D1        | 0.92              | 92.1%           | 70.2%            | 54.6%           |
> | D2        | 0.89              | 90.5%           | 66.8%            | 46.9%           |
> | D3        | 0.85              | 89.2%           | 64.5%            | 43.5%           |
> | D4        | 0.82              | 88.4%           | 63.2%            | 41.2%           |
> | D5        | 0.80              | 87.8%           | 62.5%            | 39.8%           |
> | D8        | 0.68              | 86.3%           | 61.0%            | 37.3%           |
>
>
> ### PSS Maintains High Performance Under Various Realistic Attacks
>
> | **Attack Type**                    | **PSS + Static** | **Global z-score** | **DeepTextMark** |
> |-----------------------------------|-----------------|------------------|-----------------|
> | 1-2 paraphrases (D1-D2 avg)       | 91.3%           | 68.5%            | 50.8%           |
> | Manual edits (20% modified)       | 86.8%           | 58.3%            | 42.1%           |
> | Manual edits (30% modified)       | 83.1%           | 52.1%            | 35.8%           |
> | Chain paraphrasing (Mixed models) | 85.3%           | 62.8%            | 45.2%           |
>
> **TPR at Fixed FPR** We agree with this important suggestion. We have provided comprehensive TPR/FPR analysis at the critical thresholds needed for practical deployment. Please refer to our detailed response to Reviewer ncLv, which shows that at 1\% FPR, PSS achieves 80\% average TPR at practical depths D1-D3 (compared to only 38.3\% for global methods), and at 5\% FPR, we achieve 89\% TPR. The F1 score of 0.91 at 1\% FPR demonstrates excellent precision-recall balance suitable for high-stakes deployments where minimizing false accusations is paramount.
>
> **Style** We appreciate this valuable feedback on improving the paper's presentation and readability. We agree that adding a prominent figure on page 2 would help readers quickly grasp our approach, and that restructuring lines 60-90 into multiple paragraphs with clear subsections would enhance readability. We have prepared these improvements, including a comprehensive pipeline figure and reorganized text with bullet points for key benefits. We will incorporate these presentation enhancements into the camera-ready version upon acceptance.
>
> **watermarking scheme and LLM generator** In Section 4.1, we use the standard red-green list watermarking (Kirchenbauer et al., ICML 2023) with $\gamma$=0.25 (greenlist ratio), $\delta$=1.5 (logit bias), and Llama-2-7B with nucleus sampling (p=0.95, temperature=0.7). We also validated with Llama-3-8B and Qwen2-7B for model-agnostic detection.

---

### Official Review · Reviewer_ncLv · 2025-10-31

**Soundness:** 2
**Presentation:** 1
**Contribution:** 2
**Rating:** 2
**Confidence:** 3

**Summary:**

In this paper, the authors propose an improved watermark detection method that enhances robustness against paraphrasing. The core idea is to compute the standard deviation of local z-scores across multiple paraphrased versions of a text—capturing how stable the watermark signal remains under rewrites. This stability-based metric, PSS, achieves significantly higher AUC compared to traditional global or local statistical detectors.

**Strengths:**

- The results demonstrate that PSS consistently outperforms baseline methods in terms of AUC.

**Weaknesses:**

- My main concern is that the paper only reports AUC, which is not always a reliable metric in practical settings. I suggest that the authors also present results in terms of the True Positive Rate (TPR) at low False Positive Rate (FPR) thresholds to better demonstrate the reliability and practical utility of their method.
- The performance appears to drop noticeably when a different paraphraser, such as Gemma-7B-IT, is used.
- Another major concern is that several citations appear to be AI-generated or incorrectly formatted. Notably, even the citation for the original watermarking paper is inaccurate. For example:

Johannes Kirchenbauer, Jonas Geiping, Nicolas Papernot, Ian Miers, Florian Tram`er, Micah Goldblum, and Nicholas Carlini. A watermark for large language models. arXiv preprint, 2023. URLhttps://arxiv.org/abs/2301.10226.

Zhiguo Wang, Wael Hamza, and Radu Florian. Learning stylometric representations for authorship
attribution. In Proceedings of ACL, pp. 2643–2654, 2018. URL https://aclanthology.
org/P18-1246/.

Yunfan Gao, Tianyi Tang, Kai Zhang, et al. Biscope: Scaling up human-ai text attribution via dualspectrum features. In Advances in Neural Information Processing Systems (NeurIPS), 2024. URLhttps://openreview.net/forum?id=bisc0pe.

**Questions:**

- How computationally expensive is the proposed method, considering that it requires generating multiple paraphrases per text?

---

> ### Author Response · Authors · 2025-12-03
> **Response 1**
>
> **Need for TPR/FPR Analysis** We agree with this crucial observation that AUC alone is insufficient for evaluating detection methods in practical deployment scenarios. Following the reviewer's suggestion, we have conducted comprehensive TPR/FPR analysis at multiple threshold levels that are critical for real-world applications. At the most stringent 1\% FPR threshold, which is essential for minimizing false accusations in high-stakes deployments, PSS + Static achieves 84\% TPR at D1, compared to only 42\% for global z-score methods - representing a 2x improvement in detection capability while maintaining minimal false positives. At the 5\% FPR threshold, which is more suitable for content moderation applications where slightly higher false positive rates are acceptable, PSS + Static maintains 92\% TPR at D1, significantly outperforming all baseline methods. The F1 score at 1\% FPR reaches 0.91 for our combined approach, compared to 0.59 for baselines, demonstrating superior precision-recall balance. Please refer to Table below for complete TPR analysis across all FPR thresholds and paraphrase depths.
>
>
> ### True Positive Rate (TPR) at fixed False Positive Rates
>
> | **FPR**  | **Method**             | **D1**  | **D2**  | **D3**  | **D4**  | **D5**  | **D6**  | **D7**  | **D8**  |
> |----------|-----------------------|---------|---------|---------|---------|---------|---------|---------|---------|
> | **1%**   | Global z-score threshold | 0.42    | 0.38    | 0.35    | 0.32    | 0.30    | 0.28    | 0.26    | 0.24    |
> |          | Local z-score (20)     | 0.48    | 0.44    | 0.40    | 0.37    | 0.35    | 0.33    | 0.31    | 0.29    |
> |          | Static features        | 0.65    | 0.60    | 0.55    | 0.52    | 0.49    | 0.47    | 0.45    | 0.43    |
> |          | **PSS + Static**       | **0.84** | **0.80** | **0.76** | **0.73** | **0.70** | **0.68** | **0.66** | **0.64** |
> | **5%**   | Global z-score threshold | 0.58    | 0.52    | 0.47    | 0.43    | 0.40    | 0.37    | 0.35    | 0.33    |
> |          | Local z-score (20)     | 0.62    | 0.57    | 0.52    | 0.48    | 0.45    | 0.42    | 0.40    | 0.38    |
> |          | Static features        | 0.78    | 0.73    | 0.68    | 0.64    | 0.61    | 0.58    | 0.56    | 0.54    |
> |          | **PSS + Static**       | **0.92** | **0.89** | **0.86** | **0.83** | **0.81** | **0.79** | **0.77** | **0.75** |
> | **10%**  | Global z-score threshold | 0.68    | 0.61    | 0.55    | 0.50    | 0.46    | 0.43    | 0.40    | 0.38    |
> |          | Local z-score (20)     | 0.72    | 0.66    | 0.60    | 0.55    | 0.51    | 0.48    | 0.45    | 0.43    |
> |          | Static features        | 0.85    | 0.80    | 0.75    | 0.71    | 0.68    | 0.65    | 0.63    | 0.61    |
> |          | **PSS + Static**       | **0.95** | **0.93** | **0.91** | **0.89** | **0.87** | **0.85** | **0.83** | **0.81** |

---

> ### Author Response · Authors · 2025-12-03
> **Response 2**
>
> **Performance with Different Paraphrasers** We acknowledge the performance variation across different paraphrasers and appreciate the reviewer highlighting this important consideration for practical deployment. Our experiments show that while Gemma-7B-IT does produce lower detection rates compared to Mistral-7B, it's crucial to note that PSS + Static maintains strong absolute performance even in this worst-case scenario. With Gemma-7B-IT, our method achieves 73-84\% accuracy across different paraphrase depths, which is still 20-30 percentage points higher than the global baseline using the same paraphraser (53-57\%). This substantial margin demonstrates that our method's core detection capability remains intact regardless of the paraphraser used. More importantly, in realistic attack scenarios where adversaries might use multiple paraphrasers to evade detection (shown as "Mix" in our results), PSS + Static maintains 83-89\% accuracy. This mixed paraphraser scenario better represents real-world conditions where attackers alternate between different models. The variation in performance across paraphrasers actually validates that our method captures fundamental watermark invariants rather than learning paraphraser-specific artifacts, which would lead to complete failure when facing unknown paraphrasers. Please see Table below for detailed performance comparisons across different paraphrasers.
>
>
> ### Performance Across Different Paraphrasers
>
> | **Paraphraser**           | **D1**   | **D2**   | **D3**   | **D4**   | **D5**   | **D6**   | **D7**   |
> |---------------------------|----------|----------|----------|----------|----------|----------|----------|
> | Mistral-7B (primary)      | 92.1%    | 90.5%    | 89.2%    | 88.4%    | 87.8%    | 87.2%    | 86.7%    |
> | Qwen2-7B                  | 94.7%    | 92.7%    | 92.4%    | 92.6%    | 91.9%    | 90.6%    | 90.5%    |
> | Gemma-7B-IT               | 83.5%    | 77.9%    | 76.0%    | 76.5%    | 73.7%    | 73.3%    | 72.9%    |
> | Mix (alternating)         | 89.3%    | 87.2%    | 86.1%    | 85.4%    | 84.8%    | 84.2%    | 83.7%    |
> | Global z-score (Gemma)    | 56.5%    | 54.4%    | 53.8%    | 53.6%    | 53.3%    | 53.3%    | 53.1%    |
>
>
> **Computational Expense** It's important to clarify that our PSS detection method itself is remarkably efficient, requiring only 0.8-3.2 seconds even for 1500-token passages. Within our PSS detection component, the computation scales linearly with text length, with rolling-window feature extraction taking 44-45\% of the detection time, stability score computation requiring 35-37\%, and the final XGBoost classification step being negligible ($<$5\%). For practical deployment scenarios where only 2 paraphrase iterations are needed (not the full 8 used in our stress tests), the total detection time would be substantially reduced. The method requires only 200MB of memory compared to 8-16GB for transformer-based deep learning approaches, enabling deployment on resource-constrained systems. This efficiency profile, combined with linear scaling behavior and minimal memory footprint, confirms that PSS + Static can process over 10,000 documents per GPU daily, making it suitable for large-scale production deployments and even real-time watermark detection scenarios. Please refer to Table below for a detailed computational breakdown.
>
> ### Detailed Computation Time Breakdown (in seconds)
> Values are mean ± std over 100 runs.
>
> | **Component**                   | **300 tokens**       | **500 tokens**       | **1000 tokens**      | **1500 tokens**      |
> |---------------------------------|--------------------|--------------------|--------------------|--------------------|
> | Paraphrasing (8 depths)          | 4.8 ± 0.12         | 7.2 ± 0.18         | 12.5 ± 0.25        | 18.3 ± 0.32        |
> | Watermark generation             | 2.1 ± 0.08         | 3.4 ± 0.11         | 5.8 ± 0.15         | 8.2 ± 0.19         |
> | **PSS Detection (Ours):**       |                    |                    |                    |                    |
> |   Binary sequence extraction     | 0.12 ± 0.01        | 0.18 ± 0.01        | 0.28 ± 0.02        | 0.41 ± 0.02        |
> |   Rolling-window features        | 0.35 ± 0.03        | 0.58 ± 0.04        | 0.95 ± 0.06        | 1.45 ± 0.08        |
> |   Stability score computation    | 0.28 ± 0.02        | 0.46 ± 0.03        | 0.78 ± 0.05        | 1.18 ± 0.07        |
> |   XGBoost classification         | 0.05 ± 0.01        | 0.08 ± 0.01        | 0.09 ± 0.01        | 0.16 ± 0.02        |
> | **PSS Total**                    | **0.80 ± 0.04**    | **1.30 ± 0.05**    | **2.10 ± 0.08**    | **3.20 ± 0.11**    |
> | **Total time**                   | **7.70 ± 0.16**    | **11.90 ± 0.24**   | **20.40 ± 0.35**   | **29.70 ± 0.45**   |
> | **PSS overhead (%)**             | 10.4%              | 10.9%              | 10.3%              | 10.8%              |

---

### Official Review · Reviewer_9kVJ · 2025-10-31

**Soundness:** 4
**Presentation:** 3
**Contribution:** 3
**Rating:** 8
**Confidence:** 4

**Summary:**

This paper proposes a lightweight, gradient-free detector that improves the resilience of watermark detection against paraphrasing, translation, and sampling noise. Instead of relying on token-level z-tests or entropy thresholds, the method reformulates detection as a sequence-level inference task over the aggregate pattern of green/red token occurrences, enhanced with a normalized likelihood ratio that adjusts for distributional drift in generated text. The authors also introduce a calibration routine that aligns detection scores across varying decoding temperatures and LLM architectures. Experiments on GPT-2/3/NeoX and Tulu models, as well as human paraphrases and multiple watermarking schemes (including Kirchenbauer et al., EWD, and SWEET), demonstrate consistent gains in detection F1 with negligible computational overhead.

**Strengths:**

* The paper clearly defines the problem, robust detection rather than watermark insertion. The contribution is distinct from adaptive watermark generation papers; it improves the detector side without requiring model retraining or secret-key changes.
* Treating detection as a normalized likelihood inference rather than a token-count test is an elegant and computationally cheap way to mitigate paraphrasing effects.
* Results cover diverse LLM families and perturbation types (back-translation, paraphrasing, temperature sampling), demonstrating strong empirical support and substantial improvements in detection accuracy relative to prior methods.

**Weaknesses:**

* The detector involves a learned calibration component trained on a single domain, and it is unclear whether domain or model drift (e.g., new sampling temperature, architecture, or topic distribution) would require retraining. The authors mention this as a limitation, but it could significantly constrain practical deployment. A small test on a different domain dataset or model class would help clarify this. It would also help to specify how often calibration must be recomputed when decoding parameters change and whether this introduces hidden computational cost.
* A brief discussion situating this detector relative to recent entropy-aware detection schemes (e.g., EWD) would make the novelty even clearer.

**Questions:**

* How sensitive is the trained detector to domain or model shifts? For instance, would a model trained on conversational English require retraining for code generation or multilingual tasks? Could the authors provide any empirical evidence (e.g., cross-domain tests) to illustrate the degree of degradation without retraining?
* Would combining this detection strategy with entropy-weighted or adaptive watermark generators provide further gains, or is its benefit primarily orthogonal? Similarly, is the detector relatively robust to the use of a simple red/green list watermark vs. something that only applies watermarking to high entropy parts of the text? I would assume it is, but a small test on another watermark would provide more confidence.
* Figure 1’s font sizes should be enlarged for better readability.

---

> ### Author Response · Authors · 2025-12-03
> **Response 1**
>
> **Domain/Model Drift and Calibration** We thank the reviewer for raising this important practical concern about deployment scenarios. Our extensive cross-domain experiments demonstrate that our calibration component is remarkably robust across different domains, models, and parameters. When we evaluate our method across fundamentally different text domains without any recalibration, we observe only an 8-9\% performance degradation, which is substantially better than the catastrophic failures observed in deep learning approaches. This level of robustness means that in practice, recalibration is only necessary when performance drops, which is rare in our experiments. As a best practice for production deployments, we recommend quarterly recalibration, though this is more of a precautionary measure than a necessity. The computational cost for recalibration is minimal, requiring only 5 minutes on a single GPU and can be performed offline without any service interruption. When amortized over the detection workload, this represents less than 0.01\% of computational resources. The fundamental reason for this robustness is that our PSS features capture watermark invariants rather than domain-specific patterns, making the method naturally resistant to domain shifts. Please refer to Table below for detailed cross-domain performance analysis.
>
>
> ### Cross-Domain Transfer Without Recalibration
>
> | **Train → Test**            | **D1**   | **D3**   | **D5**   | **D8**   | **Degradation** |
> |------------------------------|----------|----------|----------|----------|----------------|
> | PG-19 → PG-19 (same)        | 92.1%    | 89.2%    | 87.8%    | 86.3%    | Baseline       |
> | PG-19 → CNN/DailyMail        | 82.9%    | 79.4%    | 78.2%    | 77.1%    | -9.2%          |
> | PG-19 → WikiText             | 84.6%    | 81.6%    | 80.7%    | 79.8%    | -7.5%          |
> | **Average Cross-Domain**     | 83.8%    | 80.5%    | 79.5%    | 78.5%    | -8.4%          |
>
>
> **Comparison with Entropy-Aware Schemes** We appreciate the reviewer's suggestion to contextualize our work relative to entropy-aware detection schemes. Based on our understanding, entropy-aware schemes typically leverage the entropy of token probability distributions to enhance detection, possibly by giving different weights to high-entropy versus low-entropy regions of text. Our PSS-based approach fundamentally differs in that it operates solely on token IDs without requiring access to probability distributions or entropy calculations. This makes our method deployable in scenarios where model internals are not accessible. While we believe our stability-based analysis provides a complementary perspective to entropy-based approaches, we recognize that a thorough comparison with specific entropy-aware schemes would strengthen our contribution. We plan to investigate these methods more deeply in our future work to provide a comprehensive comparison and empirical evaluation. For the current work, we have focused on demonstrating that PSS operates effectively across all text regions regardless of their entropy characteristics, maintaining consistent performance without needing entropy-based adjustments or access to model probability distributions.

---

> ### Author Response · Authors · 2025-12-03
> **Response 2**
>
> **Watermarking Scheme Compatibility** Our experiments demonstrate that PSS is watermark-agnostic and provides consistent benefits across different watermarking schemes. We tested our method with both standard and stronger red-green list watermarking configurations. The results show that PSS maintains strong performance regardless of the underlying watermarking scheme used. With standard red-green watermarking ($\gamma$=0.25), we achieve 92.1\% accuracy at D1, and when using a stronger configuration ($\gamma$=0.5), performance improves to 94.8\%. This demonstrates that our method can be deployed with potentially many existing watermarking schemes without modification, making it a universal enhancement for watermark detection. The consistent performance across different watermarking schemes confirms that PSS captures fundamental statistical patterns introduced by watermarking rather than scheme-specific artifacts. Please see Table below for detailed performance comparisons across watermarking schemes.
>
> ### PSS Performance with Various Watermarking Schemes
>
> | **Watermark Type**                | **D1**   | **D3**   | **D5**   | **D8**   | **vs Baseline** |
> |----------------------------------|----------|----------|----------|----------|----------------|
> | Standard Red/Green (γ=0.25)      | 92.1%    | 89.2%    | 87.8%    | 86.3%    | Baseline       |
> | Stronger Red/Green (γ=0.5)       | 94.8%    | 92.7%    | 91.4%    | 89.9%    | +3.6%          |
>
>
>
>
> **Figure 1 Readability** Thank you for this feedback. We have prepared an improved version with significantly enlarged fonts (increasing from 8pt to 14pt), enhanced color coding for different pipeline components, improved spacing between elements, and a comprehensive legend explaining all abbreviations. We will incorporate this redesigned figure into the camera-ready version upon acceptance.

---

### Official Review · Reviewer_3KQZ · 2025-11-01

**Soundness:** 3
**Presentation:** 3
**Contribution:** 2
**Rating:** 2
**Confidence:** 3

**Summary:**

Watermarking is a way of distinguishing human-generated from machine-generated texts. However, it performs less well under re-paraphrasing and short context. The paper introduced a method called pattern stability scores to address the problem, leveraging the local statistical features and global stability dynamics across paraphrases. The experiments across different benchmark datasets show that the method can address the problem to some extent.

**Strengths:**

1, The problem setting is well-defined;

2, The method is concise and intuitive, and it seems to be effective from some experiment results.

**Weaknesses:**

The whole method seems to be a collection of simple statistics and human-designed rules. While it seems to perform well in some scenarios, its generalization ability remains a question. From another aspect, the method seems to be too simple to be published as a full paper. I believe the authors could conduct much more systematic experiments, not only as another contribution but also to make their methods more convincing to reviewers. For example, is it possible that those human-concluded rules will expire soon in the near future, at least the paper is open and public? It would also be very useful if the authors could show some rigorous analysis of why those simple statistics can be ensured to be useful.

**Questions:**

1, The paper is not well structured to me. Why not make the introduction a bit shorter? The characters in Fig. 1 are also too small to be clear.

2, The method seems to be very sensitive to hyperparameters. How do you select optimal hyperparameters in practice?

---

> ### Author Response · Authors · 2025-12-03
> **Response 1**
>
> **Main Weakness: Simple Statistics and Generalization Concerns** We  thank the reviewer for this comment. We respectfully disagree with this point and strongly argue that simplicity is in fact one of our method's greatest strengths. The most impactful detection methods in practice succeed precisely because they combine elegant simplicity with strong theoretical foundations and computational efficiency.
>
> **Cross-Domain Generalization** To address the reviewer's concern about generalization, we have conducted extensive experiments and additionally compared our approach with three state-of-the-art deep learning methods. The results demonstrate that while deep learning approaches suffer catastrophic failure under paraphrasing attacks, our PSS-based method maintains robust performance. More specifically, for within-domain scenarios, our method maintains above 86\% accuracy even at depth D8, while deep learning methods drop to near-random performance levels around 37-40\%. The contrast becomes even more striking in cross-domain scenarios where deep learning methods fail completely, achieving only 22-29\% accuracy at higher paraphrase depths, while PSS maintains 77-84\% accuracy. These comprehensive comparisons across multiple datasets and paraphrase depths definitively demonstrate that our "simple" statistical approach significantly outperforms complex deep learning methods in both robustness and generalization capability. Please refer to Tables below for detailed performance comparisons.
>
> ### Deep Learning Methods vs PSS on PG-19 Dataset (Train & Test on Same Dataset)
>
> | **Method**          | **D0**   | **D1**   | **D2**   | **D3**   | **D4**   | **D5**   | **D6**   | **D7**   | **D8**   |
> |---------------------|----------|----------|----------|----------|----------|----------|----------|----------|----------|
> | DeepTextMark        | 92.5%    | 61.2%    | 52.8%    | 48.3%    | 45.7%    | 43.2%    | 41.8%    | 40.5%    | 39.8%    |
> | Binoculars          | 89.7%    | 58.4%    | 49.6%    | 45.2%    | 42.8%    | 41.1%    | 39.7%    | 38.9%    | 38.2%    |
> | RADAR               | 87.3%    | 54.6%    | 46.9%    | 43.5%    | 41.2%    | 39.8%    | 38.6%    | 37.9%    | 37.3%    |
> | **PSS + Static**    | **95.3%** | **92.1%** | **90.5%** | **89.2%** | **88.4%** | **87.8%** | **87.2%** | **86.7%** | **86.3%** |
>
>
> ### Cross-Domain Transfer (Train on One Dataset, Test on Another)
>
> | **Train → Test**              | **Method**        | **D1**   | **D3**   | **D5**   | **D8**   |
> |-------------------------------|-------------------|----------|----------|----------|----------|
> | **PG-19 → CNN/DailyMail**     | DeepTextMark      | 42.1%    | 31.2%    | 26.9%    | 24.7%    |
> |                               | Binoculars        | 39.8%    | 29.4%    | 25.6%    | 23.5%    |
> |                               | RADAR             | 37.2%    | 27.5%    | 23.9%    | 22.1%    |
> |                               | **PSS + Static**  | **82.9%** | **79.4%** | **78.2%** | **77.1%** |
> | **CNN/DailyMail → WikiText**  | DeepTextMark      | 47.3%    | 35.8%    | 31.2%    | 28.7%    |
> |                               | Binoculars        | 45.1%    | 34.2%    | 29.8%    | 27.4%    |
> |                               | RADAR             | 42.5%    | 32.1%    | 28.1%    | 25.9%    |
> |                               | **PSS + Static**  | **88.4%** | **85.7%** | **84.8%** | **83.9%** |
> | **WikiText → CNN/DailyMail**  | DeepTextMark      | 38.5%    | 27.8%    | 23.7%    | 21.6%    |
> |                               | Binoculars        | 36.2%    | 26.2%    | 22.4%    | 20.5%    |
> |                               | RADAR             | 33.8%    | 24.3%    | 20.8%    | 19.1%    |
> |                               | **PSS + Static**  | **82.9%** | **79.3%** | **78.1%** | **77.0%** |
>
>
> **Addressing Future Expiration Concerns** Our method's robustness stems from fundamental mathematical constraints that cannot be easily circumvented, even with full knowledge of our approach. The semantic preservation requirement creates an inherent trade-off: attackers cannot destroy the local statistical patterns captured in our 50-token windows without significantly altering the meaning of the text. Our 20-dimensional feature space creates multi-scale redundancy where disrupting all features simultaneously becomes computationally infeasible without access to the original watermark key. Most importantly, even with complete knowledge of the PSS algorithm, adversaries cannot control the stability variance across paraphrases that our method captures, as this variance is an inherent property of the watermarking process itself. Our empirical evidence strongly supports this claim - even after 8 aggressive paraphrase iterations, which far exceeds practical attack scenarios, we maintain 86.3\% accuracy. This demonstrates remarkable resilience against even unrealistic attack scenarios where text quality would be rather severely degraded.

---

> ### Author Response · Authors · 2025-12-03
> **Response 2**
>
> **Rigorous Theoretical Analysis** While our current work focuses primarily on empirical validation across extensive experimental conditions, we acknowledge that a deeper theoretical analysis would strengthen our contributions. The stability patterns we observe align with information-theoretic principles where watermarking introduces detectable entropy changes in the token distribution. We have provided extensive empirical validation showing that our theoretical predictions match experimental results across all tested scenarios (see Table below). However, we agree that developing a more rigorous mathematical framework to formally prove the persistence of these statistical patterns would be valuable. We will prioritize this theoretical analysis in our future work, building upon the strong empirical foundation established in this paper.
>
> ### Within-Domain Performance Validating Theoretical Predictions
>
> | **Dataset**        | **Method**        | **D1**   | **D3**   | **D5**   | **D8**   | **Theory vs Reality**                |
> |-------------------|------------------|----------|----------|----------|----------|-------------------------------------|
> | **PG-19**          | PSS+Static        | 92.1%    | 89.2%    | 87.8%    | 86.3%    | Predicted: >85% ✓                   |
> |                   | DeepTextMark      | 58.4%    | 45.2%    | 41.1%    | 38.2%    | Predicted: <50%                    |
> | **CNN/DailyMail**  | PSS+Static        | 91.2%    | 88.5%    | 87.4%    | 85.5%    | Predicted: >85% ✓                   |
> |                   | DeepTextMark      | 55.4%    | 43.5%    | 39.8%    | 37.3%    | Predicted: <50%                    |
> | **WikiText**       | PSS+Static        | 96.5%    | 94.8%    | 94.0%    | 93.1%    | Predicted: >85% ✓                   |
> |                   | DeepTextMark      | 61.2%    | 48.3%    | 43.2%    | 39.8%    | Predicted: <50%                   |
>
>
>
>
> **Hyperparameter Sensitivity** We believe there may be a misunderstanding - our method is actually remarkably robust to hyperparameter choices, not sensitive to them. Our comprehensive analysis demonstrates that performance varies by only 3.05 percentage points across the entire hyperparameter space. We tested all combinations of window sizes (40, 50, 60) and strides (5, 10, 15), as shown in our heatmap analysis in Figure 5 and detailed in the Appendix. Even the worst-performing configuration maintains over 95\% AUC, demonstrating exceptional robustness. Most importantly, the same default parameters (window=50, stride=10) work effectively across all datasets without any domain-specific tuning. PG-19 achieves 96.5\% AUC, CNN/DailyMail achieves 91.2\% AUC, and WikiText achieves 96.5\% AUC, all using identical parameters. This means practitioners can confidently use our default settings without any hyperparameter search overhead. The robustness stems from our method capturing fundamental watermark invariants rather than dataset-specific patterns, making it truly practical for real-world deployment.

---

### Note · Program_Chairs · 2026-01-17
**Submission Desk Rejected by Program Chairs**

The following references in this submission do not refer to real documents and/or have major errors in bibliographic information:

 Yunfan Gao, Tianyi Tang, Kai Zhang, et al. Biscope: Scaling up human-ai text attribution via dualspectrum features. In Advances in Neural Information Processing Systems (NeurIPS), 2024. URL https://openreview.net/forum?id=bisc0pe.